# Presence of Dendritic Cell Subsets in Sentinel Nodes of Breast Cancer Patients Is Related to Nodal Burden

**DOI:** 10.3390/ijms23158461

**Published:** 2022-07-30

**Authors:** Joanna Szpor, Joanna Streb, Anna Glajcar, Piotr Sadowski, Anna Streb-Smoleń, Robert Jach, Diana Hodorowicz-Zaniewska

**Affiliations:** 1Department of Pathomorphology, Jagiellonian University Medical College, 31-008 Cracow, Poland; joanna.szpor@uj.edu.pl (J.S.); glajcar.anna@gmail.com (A.G.); piotr.sadowski@uj.edu.pl (P.S.); 2Department of Oncology, Jagiellonian University Medical College, 31-008 Cracow, Poland; 3Department of Oncology, Maria Sklodowska-Curie National Research Institute of Oncology, 00-001 Cracow, Poland; aniastreb@gmail.com; 4Department of Gynecology and Oncology, Jagiellonian University Medical College, 31-008 Cracow, Poland; robert.jach@uj.edu.pl; 5General, Oncological and Gastrointestinal Surgery, Jagiellonian University Medical College, 31-008 Cracow, Poland; diana.hodorowicz-zaniewska@uj.edu.pl

**Keywords:** breast cancer, dendritic cells, lymph nodes, lymphatic metastases

## Abstract

BACKGROUND: Sentinel lymph nodes (SLNs) are both the first site where breast cancer (BC) metastases form and where anti-tumoral immunity develops. Despite being the most potent antigen-presenting cells, dendritic cells (DCs) located in a nodal tissue can both promote or suppress immune response against cancer in SLNs. METHODS: In SLNs excisions obtained from 123 invasive BC patients, we performed immunohistochemistry (IHC) for CD1a, CD1c, DC-LAMP, and DC-SIGN to identify different DCs populations. Then we investigated the numbers of DCs subsets in tumor-free, micrometastatic, and macrometastatic SLNs with the use of a light microscope. RESULTS: We observed that CD1c+ and DC-SIGN+ DCs were more numerous in SLNs with a larger tumor size. More abundant intratumoral DC-LAMP+ population was related to a higher number of metastatic lymph nodes. Conversely, more abundant CD1a+ DCs were associated with a decreasing nodal burden in SLNs and a lower number of involved lymph nodes. Moreover, densities of the investigated DC populations differed with respect to tumor grade, HER2 overexpression, hormone receptor status, and histologic type of BC. CONCLUSIONS: According to their subtype, DCs are associated with either lower or higher nodal burden in SLNs from invasive BC patients. These relationships appear to be dependent not only on the maturation state of DCs but also on the histological and biological characteristics of the tumor.

## 1. Introduction

Metastases in lymph nodes are one of the most important adverse prognostic indicators in invasive breast cancer (BC) patients [1]. Sentinel lymph nodes (SLNs) are the first lymphatic organs on the lymph flow from a primary tumor; therefore, they are regarded as sites where regional breast cancer metastastes occur first [1,2,3]. Recently, sentinel lymph node biopsy (SLNB) has displaced axillary lymph node dissection (ALND) in the management of primary operable and clinically node-negative BC, as it provides comparable staging information but spares many side effects associated with ALND without deterioration in patient survival rates [1,4]. Evaluation of nodal burden in SLNs is based on the size of the largest metastatic focus, which is assigned to one of three categories: isolated tumor cells (tumor diameter < 0.2 mm or < 100 cancer cells), micrometastasis (0.2–2 mm), or macrometastasis (>2 mm) [1,3]. The diameter of secondary tumor correlates negatively with patient survival rates [3].

Despite being the most likely location of BC spread, SLNs are the sites where the anti-cancer immune response develops. This process is initiated by dendritic cells (DCs), the most potent antigen-presenting cells. DCs loaded with tumor-derived molecules migrate from the primary tumor site to SLNs, where they are able to present antigens and subsequently activate T-cells [2,3,5]. The notion of SLNs as the site of activation of antitumoral immune response in BC was postulated by Poindexter et al, who observed higher levels of interleukin (IL) 10 and IL-12 in SLNs than in other uninvolved lymph nodes [5]. Higher numbers of DCs in tumor-free and metastatic cervical and endometrial cancer SLNs as compared to their non-sentinel counterparts backed up the hypothesis of SLNs as more immunologically active organs than other LNs [6]. However, anti-tumor immune reaction in SLNs is also modulated by factors produced by tumor cells and their microenvironment; this includes impairment of antigen presentation by DCs and effect on their maturation status [2,3,5]. When immature, the main function of DCs is antigen processing and maintenance of immune tolerance. After a contact with antigen and additional stimulation, DCs maturate, migrate from the peripheral tissue to lymphoid organs, and acquire the ability to induce immune response [7]. It was shown that maturation of DCs may be in part dependent on VEGF expression by breast cancer cells [8]. However, a study carried out in melanoma cases showed that even mature DCs retain their antigen-processing functions [9].

To date, several subtypes of DCs have been distinguished. Although there is no universal marker for DC identification, functional and maturation status of DCs is determined by the set of expression of certain proteins [7] that allow approximate classification of DC subtypes. Mature DCs in cooperation with T helper and cytotoxic cells were shown to contribute to cancer cell killing. The proposed role of DCs in this process was supply of cytokines and other “contact signals” that launch anti-tumoral immunity [10]. On the other hand, prevalence of DC subtypes associated with humoral immune response and immunological tolerance was observed in LNs from non-small cell lung cancer patients and suggested as one of the mechanisms of tumor-escape. Moreover, reduction of DCs in patient blood (as compared with healthy donors) pointed to the systemic impact of cancer-derived molecules on DC number [11]. Therefore, immune suppression is not confined to the primary tumor site, but may extend to the circulation and secondary lymphoid organs also [12]. It was suggested that immune suppression in SLNs may precede the establishment of nodal metastases [2] and that immunosuppression follows anti-tumoral response (regarded as maturation of DCs and activation of specific T-cells) in BC [13,14,15]. Thus, it is not surprising that DCs are considered as one of the targets of immunotherapeutic approaches for BC patients [2,7,13,15].

Recently we have shown that the content of respective DCs subtypes in BC tumors differs according to BC molecular subtype and clinicopathological features [16]. In this study, we investigated the densities of several DC populations in SLNs of invasive breast cancer patients with respect to the size of nodal metastasis and other prognostic parameters in this malignancy.

## 2. Results

Clinicopathological characterization of the study group is summarized in Table 1.

### 2.1. Relationships between Densities of DC Populations and Metastatic Burden of SLNs

When micro- and macrometastatic SLNs were compared, we found significantly higher densities of CD1c+ and DC-SIGN+ DCs at tumor margin of the latter (U Mann–Whitney test: *p* < 0.003 and *p* < 0.035, respectively; Figure 1B,C, Table 2). Investigation of positive SLNs revealed that higher intratumoral DC-LAMP+ cell densities were associated with occurrence of metastasis in more than one LN (0.75 ± 1.38 for >1 positive SLNs vs. 0.28 ± 1.04 for 1 positive SLN, *p* < 0.015; Figure 1E). With regard to type of metastasis in SLN, we observed that CD1a+ DCs tended to decrease their numbers in distant area from tumor-free SLNs, SLNs with micrometastases to SLNs with macrometastasis (*p* = 0.055; Figure 1A, Table 2). Moreover, there was a tendency towards more abundant CD1a+ DCs in distant area of tumor-free than in positive SLNs (63.25 ± 31.16 vs. 50.85 ± 29.46, *p* = 0.054, Figure 1D). The intratumoral area of macrometastases showed the most abundant infiltration of CD1a+, followed by DC-LAMP+, CD1c+, and DC-SIGN+ DCs (Figure 2, Table 2).

We found that intratumoral DC-LAMP+ DCs as well as DC-LAMP+ and CD1c+ cells located at the tumor edge correlated positively with the size of metastatic tumor (R = 0.28, *p* < 0.035; R = 0.39, *p* < 0.002 and R = 0.27, *p* < 0.035, respectively). CD1a+ DCs located in the distant area of SLN showed negative correlations (R = −0.27, *p* < 0.007) with the number of positive lymph nodes. Conversely, intratumoral DC-LAMP+ cells correlated positively (R = 0.35, *p* < 0.007) with this parameter.

### 2.2. Differences in Densities of DC Populations in SLNs and Other Prognostic Indicators in BC

Densities of CD1a+ cells in distant area of SLNs were significantly higher in G1 (66.53 ± 19.46, *p* < 0.020) and G2 (63.48 ± 32.26, *p* < 0.035) than in G3 cancers (45.60 ± 30.12). Moreover, DC-LAMP+ DCs of intratumoral and distant area tended to decrease their numbers in cancers of higher grade (G1: 1.23 ± 2.00, G2: 0.72 ± 1.47, G3: 0.10 ± 0.33, *p* = 0.067 and G1: 151.46 ± 46.90, G2: 152.21 ± 55.59, G3: 126.35 ± 56.94, *p* = 0.078, respectively). With respect to HR status, we found higher densities of CD1c+ cells in the distant node area in HR-positive cancers than in HR-negative (51.01 ± 35.72 vs. 24.47 ± 28.73, *p* < 0.025). A similar tendency was also observed for the CD1c+ population of tumor border (HR-positive BC: 24.13 ± 23.79, HR-negative BC: 4.25 ± 7.98, *p* = 0.057). With reference to HER2 status, we noted tendencies for lower CD1a+ DCs and higher CD1c+ cells in distant area of SLNs from HER2-overexpressing cases than in tissues with normal HER2 expression (44.72 ± 23.71 vs. 58.61 ± 23.71, *p* = 0.058 and 60.68 ± 37.45 vs. 44.41 ± 34.92, *p* = 0.064, respectively). For histologic type, significantly more numerous CD1a+ cells were found in the intratumoral area of CLI as compared with NOS tumors (2.75 ± 1.00 vs. 1.37 ± 2.08, *p* < 0.040).

### 2.3. Correlations between Densities of DC Populations

The densities of DC-SIGN+ cells at tumor border correlated with DC-LAMP+ cells (R = 0.41, *p* < 0.004) and CD1c+ cells (R = 0.41, *p* < 0.005) at the tumor edge as well as with intratumoral CD1c+ population (R = 0.33, *p* < 0.035). For DC-SIGN+ population of the distant area, correlations with CD1c+ cells located in intratumoral (R = 0.39, *p* < 0.015) and distant (R = 0.27, *p* < 0.015) areas were found. CD1a+ cells at tumor edge correlated with DC-LAMP+ (R = 0.64, *p* < 0.001) and CD1c+ cells (R = 0.33, *p* < 0.010) at the same location. Associations between CD1a+ and DC-LAMP+ cells were observed in the distant area of SLN (R = 0.42, *p* < 0.001). Similar observations were made for DC-LAMP+ and CD1c+ cells at tumor border (R = 0.47, *p* < 0.001) as well as in the distant area (R = 0.27, *p* < 0.006).

## 3. Discussion

There is a great body of evidence that DCs are a heterogenous group of APCs with several subsets that differ in their function and role in malignant disease. The selected markers comprise different types of DCs regarding their maturational as well as functional status and can be used to approximately identify DC populations (Table 3).

CD1 encompasses a family of molecules that are responsible for the presentation of self and foreign lipid antigens on APCs. The respective members of CD1 family present a distinct repertoire of lipid antigens [26,43]. CD1a is localized mainly in early endosomes and in recycling compartments of DCs [43]. Expression of CD1a has been for a long time attributed to immature populations of DCs; however, a body of evidence showed that CD1a molecules can be found on DCs in general [2]. High expression of CD1a is a hallmark of Langerhans cells (LCs) [19]. In melanoma SLNs, CD1a+ LCs were proven to express molecules associated with mature phenotype [20]. In a study carried out by van de Ven et al., CD1a+ DCs turned out to be poor activators of T cells despite their mature phenotype. Moreover, this population in SLNs from melanoma patients was identified as skin-derived and migratory cells that eventually contribute to immune tolerance [18]. Accumulation of immature CD1a+ DCs in metastatic oral squamous cell carcinoma (OSCC) LNs was associated with both increase in regulatory T cells and a drop of cytotoxic cell numbers, supporting the hypothesis of the immunosuppressive microenvironment that in turn contributes to cancer spread [17].

It was shown that the cells of invasive BC attract immature CD1a+ DCs and their precursors, subsequently impair maturation of these cells, and eventually decrease their ability to activate T lymphocytes [21]. In contrast, higher densities of CD1a+ DCs both in the stroma of primary invasive tumors and in LNs were associated with absence of nodal metastases and good prognosis [22,23]. Similarly, in our recent study we showed that a higher density of intratumoral CD1a DCs was associated with longer progression-free survival in BC patients [16]. These would support the hypothesis that higher levels of DCs evoke functional anti-cancer responses [22]. Similarly, a decrease in CD1a+ cells in SLNs was associated with tumor involvement [24]. These findings are in accordance with our results, which pointed out an association between higher densities of CD1a+ DCs in lymphoid tissue and negative nodal status or lower metastatic burden in SLNs. Since such a difference was not observed by Kohrt et al. in other axillary LNs [24], it was postulated that alterations in immune profile are dependent on tumor burden in the sentinels exclusively. Furthermore, the authors suggested that changes in the immune profile occur before metastasis established and observed that decreased CD1a+ cells in non-sentinel LNs were associated with disease recurrence [24]. On the other hand, some authors did not observe any relationship between CD1a+ DCs density and metastatic status of SLNs [5,12,25].

Regarding other prognostic indicators in breast cancer, we noted that higher numbers of CD1a+ DCs in lymphoid tissue were associated with a lower tumor histological grade and HER2 negativity. This is partially in contrast with Poindexter et al.’s study, who found higher numbers of CD1a+ DCs in tumor-containing than in tumor-free SLNs from G3 cancers [5]. Correlation between CD1a level in LNs and expression of hormone receptors was also observed by some authors [22]. Although we did not make such observations, we noted that breast cancer metastases of lobular histology are more abundantly infiltrated by CD1a+ DCs than NOS cancers.

Little is known about the role of CD1c+ DCs in malignancies, particularly in BC tissues or SLNs. The CD1c molecule is expressed on Langerhans cells and some populations of B cell surfaces [26]. Inflammatory CD1c+ DCs are present in several tissues, including solid tumor infiltrate and lymph nodes [27]. In cancer patients CD1c+ cells were shown to represent DCs that are able to stimulate cytotoxic CD8+ and CD4+ helper T cells [27,28]. On the other hand, human blood CD1c+ DCs also were identified as producers of immunosuppressive and regulatory factors such as IL-10 and IDO in response to stimulation by *E. coli* bacteria [29]. CD1c+ DCs can be generated at the tumor site; their elevated numbers were noted in lung adenocarcinoma tumors as compared with normal lung tissue [30]. On the other hand, hepatocellular cancer (HCC) patient blood CD1c+ cells were identified as myeloid, IL-12 producing DCs, less abundantly represented in circulation of HCC patients than in healthy controls [31]. Similarly, lower numbers of epidermal CD1c+ DCs were found in invasive and in situ melanoma tissues in comparison with dysplastic nevi [32]. In non-small cell lung cancer, higher numbers of myeloid CD1c+ cells in tumor tissue were related to worse survival. The authors offered an explanation of tumor tissue acting like a trap, which attracts DCs but prevents their migration to LNs [11]. In our study, higher numbers of CD1c+ DCs located in close proximity to cancer islets were associated with greater diameter of metastasis as well as with HR-positive and HER2 overexpression. Thus, we propose that this DC subpopulation supports nodal spread of breast cancer and supposedly plays a more important role in progression of ER, PR, and/or HER2-positive cancers.

Lysosome-associated membrane protein (DC-LAMP) is expressed on mature DCs [2,20]. In melanoma SLNs, lower counts of mature DC-LAMP+ DCs were associated with unfavorable prognostic factors such as ulceration. As their higher numbers were related to longer survival, maturation of DCs became regarded as guarantees for long-term antimetastatic protection [35]. As opposed to this, in primary melanoma lesions, the presence of DC-LAMP+ DCs in the peritumoral area was accompanied by the infiltrate of resting naïve T-cells that pointed to a lack of T-cell stimulation [20]. High DC-LAMP+ DCs numbers of mature morphology were observed in melanoma metastatic LNs and their density in SLNs negatively correlated with additional melanoma metastases in non-SLNs. Therefore, close proximity of DC-LAMP+ DCs to T cells and tumor cells was regarded as proof of active immune reaction [34]. In tumor-free LNs from OSCC patients, numbers of DC-LAMP+ cells were higher when cancers were non-metastatic [33].

In breast cancer, more dense DC-LAMP+ cell infiltration in SLNs was associated with a lack of nodal metastases [25]. The observed numbers of these DC populations were higher in tumor-free and metastatic SLNs than in other LNs, suggesting that SLNs are sites of more potent immune activity before and early in metastasis development [25,44]. The lower proportion of mature DC-LAMP+/immature CD1a+ cells in positive breast carcinoma SLNs as compared with negative ones suggested that the maturation of DCs is arrested [25]. Moreover, lower DC-LAMP+/cytotoxic cells and higher T regs/DC-LAMP+ DCs ratios in metastatic than in negative SLNs implied impaired antigen presentation, resulting in lower cytotoxicity against tumors as well as accumulation of T regs [25]. The study using another marker of DC maturity, CD83, showed similar expression of mature DCs in tumor-negative SLNs and non-SLNs, but higher expression of co-stimulatory molecules in the latter, suggesting that immune response is suppressed before development of BC metastasis, but enhanced after the tumor is established in SLN [14]. We noted that higher infiltration of DC-LAMP+ cells in tumor islets and at tumor front is associated with more advanced nodal disease, favoring a hypothesis of insufficient or tumor-promoting immune response in BC.

Another molecule associated with DCs—DC-SIGN—is a C-type lectin receptor expressed on the surface of both immature and mature cells located in dermis and mucosal tissues [36]. Some authors associated its expression with the monocyte-derived [3] or interstitial [19] population of DCs. In primary melanoma tumors, expression of DC-SIGN and DC-LAMP was mutually exclusive [20]. This observation is in strong contrast to our results, which showed a positive correlation between these two populations in SLNs. Such an inconsistency can be explained by the notion that in peripheral tissue and in lymph nodes DC-SIGN+ represent immature and mature phenotypes of DCs, respectively [7]. However, the low expression of DC-SIGN in melanoma patient LNs suggested the loss of its expression during the maturation and migration of DCs [18]. Furthermore, we noted the positive correlation between numbers of immature CD1a+ and mature DC-LAMP+ DCs, implying distinct roles of DCs populations in lymphoid tissue or the primary tumor site. DC-SIGN is responsible for immune regulation of DCs as well as their adhesion, migration, maturation, and T cell activation. The molecule is one of the factors that contribute to immune escape of tumors [36]. DC-SIGN is also expressed on tumor-associated and monocyte-derived macrophages. Its expression can be induced by both cancer cells and fibroblasts. As a result of interaction between DC-SIGN+ macrophages and cancer cells, immunosuppressive cytokines are released from the former, which eventually promote tumor progression [38,39]. Likewise, differentiation of immunosuppressive DC-SIGN+ DCs in prostate cancer tissue was orchestrated primarily by factors derived from adjacent stroma [40]. After stimulation with LPS, DC-SIGN+ DCs were shown to dramatically increase the secretion of pro-tolerant cytokine IL-10 [37].

Information on the role of DC-SIGN+ DCs in invasive mammary tumors is modest. In primary breast tumors, immature myeloid DC-SIGN+ DCs were associated with worse survival [41] and early recurrence [42]. Since in OSCC intratumorally located immature dermal DC-SIGN+ cells were associated with worse survival, it was concluded that this DC subset fails to combat tumor effectively [33]. In primary cutaneous melanoma, immature dermal DC-SIGN DCs were increased in comparison with normal skin or naevi. In SLNs, these cells were also abundant and located primarily around high endothelial venules, although the accumulation of mature DCs was also observed [20]. In invasive BC SLNs, we observed relationship between higher DC-SIGN+ DCs infiltration of secondary tumor border and greater metastasis size, suggesting protumorigenic properties of the cells.

To sum up, we conclude that CD1a+ DCs show protective activity against cancer progression while CD1c+, DC-LAMP+ and DC-SIGN+ subsets favor tumor spread. Based on these, we might suppose that CD1a DCs are related to the better prognosis while CD1c+, DC-LAMP+, and DC-SIGN+ subsets are related to worse prognosis. Moreover, the balance between these populations can be at least partially dependent on biologic features of tumors.

## 4. Material and Methods

### 4.1. Selection of Cases

The study group consisted of primary invasive breast cancer patients who underwent tumor excision with following SLNB and/or ALND at the University Hospital in Cracow. The patients who received neoadjuvant chemotherapy were excluded from the study. The study group included patients treated at our institution between 2010 and 2020; the criteria for qualification for neoadjuvant therapy significantly changed over this period, as more patients than nowadays are treated with systemic therapy post-surgery. Our group included also 2 patients with pT4 tumors excised palliatively to avoid unstoppable bleeding.

The archival hematoxylin-eosin-stained slides were re-evaluated and one representative, well-preserved specimen was chosen for immunohistochemistry. Nottingham Histologic Grade system was used for grading, and the 8th edition of AJCC system was used for staging [45].

### 4.2. Detection of SLNs and Identification of Nodal Metastases

Lymphatic mapping was performed using the double tracer technique: the conventional blue dye method (Patent Blue V) combined with the isotope technique [46,47].

The commercially available Nanocoll^®^ kit (human albumin 500 µm/vial; GE Healthcare, Chicago, IL, USA) was used for technetium marking. The administered 99 mTc Nanocoll solution was injected subcutaneously in the perialeolar region according to protocol, 3 to 18 h prior to the surgery.

Blue dye was injected periareolarly on a table, 10 to 15 min before surgery, followed by a massage of the injection site. During surgery, a hand-held scintillation counter (Gamma Finder^®^ II) was applied to identify SLNs. Any nodes with 10% or more of the ex vivo count of the hottest node and/or any nodes with at least one blue afferent lymphatic vessels derived from the breast were removed and designated as SLNs. A successful SLNB was defined as a procedure in which at least one true sentinel node was identified to be either blue and/or radioactive.

The tumor burden of SLNs was determined histologically based on the size of the largest metastatic focus, which was assigned to one of the three categories: isolated tumor cells (tumor diameter < 0.2 mm or < 100 cancer cells), micrometastasis (0.2–2 mm), or macrometastasis (>2 mm) and the number of involved lymph nodes. When several lymph nodes were identified as sentinels, the most representative one was selected on the basis of its size and tumor content.

### 4.3. Immunohistochemistry

Immunohistochemistry (IHC) for CD1a, CD1c, DC-LAMP, DC-SIGN, estrogen receptor (ER), progesterone receptor (PR), and Ki67 protein was performed according to the protocol routinely used in our laboratory (Table 4). The selected blocks were cut into 4 μm thick sections.

UltraVision Quanto detection system (Lab Vision, ThermoScientific, Waltham, MA, USA) and 3,3′-diaminobenzidine as chromogen were used, and the slides were counterstained with Mayer hematoxylin (Thermo Fisher Scientific, Waltham, MA, USA) and coverslipped. Immunohistochemistry for HER2 (PATHWAY 4B5, Ventana Medical Systen Inc., Tucson, AZ, USA) was performed on BenchMark BMK Classic autostainer (Ventana Medical Systems Inc., Tucson, AZ, USA) using UltraVIEW DAB Detection Kit (Ventana Medical Systems Inc., Tucson, AZ, USA).

For specimens with HER2 status 2+ in immunohistochemistry, fluorescence in situ hybridization (FISH) was conducted. FISH was performed using a PathVysion HER-2 DNA Probe Kit II (Abbott Molecular, Des Plaines, IL, USA) according to the manufacturer’s protocol. The red Locus Specific Identifier (LSI) HER-2/neu and green Centromere Enumeration Probe (CEP 17) signals were counted on a fluorescence microscope equipped with specific filter sets and HER-2/neu to CEP17 ratio > 2.0 was considered as HER2/neu amplification [48].

Positive ER and PR expression thresholds were set when ≥1% of neoplastic cells showed positive immunostaining. The threshold for discriminating between low and high Ki67 expression was set at ≥20% of positive cells. Scoring of the HER2 staining was performed by a standard method [48].

### 4.4. Evaluation of DCs Densities in SLNs

First, the slides were scanned at low power (magnification of 100×) on a Nikon Labophot-2 optical microscope (Tokyo, Japan) in the search of the areas of highest infiltration of positively-stained cells. Then, DC populations were counted in 3 to 5 high-power fields (magnification of 400×, HPF) in “hot-spots” and their numbers were averaged. The densities of DCs were assessed in 3 compartments of SLN: within the islets of metastatic tumor (“intratumoral”), at the border of metastasis (“tumor margin”; no farther than 1 HPF from tumor edge), and in the “distant area”, located >1 HPF from the edge of metastasis. In tumor-free SLNs, the DC densities were evaluated in lymphoid tissue (termed “distant area”).

### 4.5. Statistical Analysis

Distributions were tested for normality with the Chi-square test. To assess the differences between groups, ANOVA Kruskal–Wallis and U Mann–Whitney tests were performed. The correlations between groups were evaluated by using the Spearman rank test. All analyses were performed using Statistica 13 (StatSoft Inc., Tulsa, OK, USA). In brackets, the mean values ± standard deviation (SD) are given; *p* values < 0.05 were considered statistically significant.

## Figures and Tables

**Figure 1 ijms-23-08461-f001:**
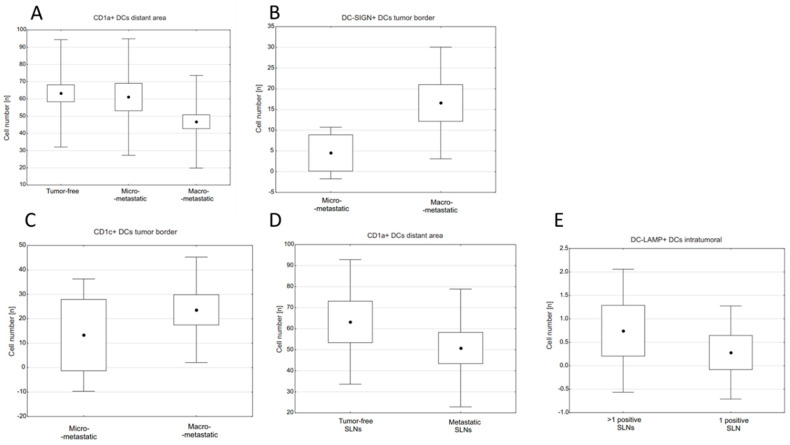
Relationships between densities of DC subpopulations and type of metastasis in SLNs or number of positive lymph nodes: (**A**) CD1a+ DC densities in the distant area of tumor-free, micrometastatic, and macrometastatic SLNs, *p* = 0.055; (**B**) DC-SIGN+ DC densities at tumor margin in micro- and macrometastatic SLNs, *p* = 0.002; (**C**) CD1c+ DC densities at tumor margin in micro- and macrometastatic SLNs, *p* = 0.031; (**D**) CD1a+ DC density in lymphoid tissue (“distant area”) with reference to presence or absence of secondary tumor in SLN, *p* = 0.054; (**E**) intratumoral DC-LAMP+ DC density with reference to presence or absence of metastases in lymph nodes other than investigated SLN, *p* = 0.013. (**A**) Kruskal–Wallis ANOVA test: the central point is arithmetical mean, box is mean ± standard error (SE) and whiskers are mean ± standard deviation (SD). (**B**–**E**) U Mann–Whitney test: the central point is arithmetical mean, box is mean ± 2xSE and whiskers are mean ± 0.95xSD. *p*-value < 0.05 was considered significant.

**Figure 2 ijms-23-08461-f002:**
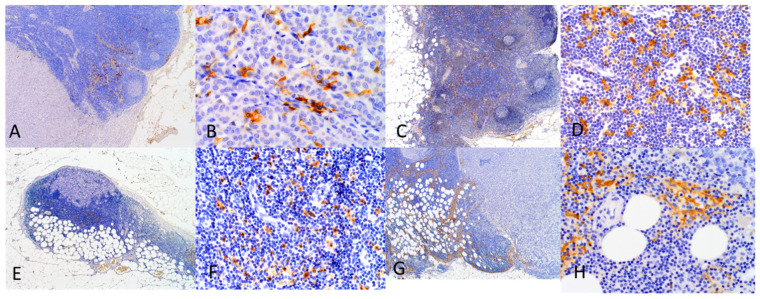
Infiltration of investigated DCs in SLNs. (**A**) CD1a+ DCs in SLN with macrometastasis. The majority of DCs are located in the distant area from the tumor and several are found at the tumor border (magnification 40×); (**B**) CD1a+ DCs observed intratumorally in BC macrometastasis (magnification 400×); (**C**) CD1c+ DCs in tumor-free SLN (magnification 40×); (**D**) CD1c+ DCs in tumor-free SLN (magnification 400×); (**E**) DC-LAMP+ DCs in SLN with micrometastasis. The DCs are located predominantly at the tumor border (magnification 40×); (**F**) DC-LAMP+ DCs observed in the area distant from micrometastasis (400×); (**G**) DC-SIGN+ DCs in SLN with macrometastasis. The DCs are located at the tumor border and in a distant area of the tumor (magnification 40×). (**H**) DC-SIGN+ DCs observed at the tumor border of macrometastasis (400×).

**Table 1 ijms-23-08461-t001:** Clinicopathological characteristics of the study group.

Characteristic	Number of Cases (Total = 123)	%
Age (years):		
Range	29–87	
Mean	55	
Nodal burden:		
Tumor-free	43	35
Micrometastases	22	17.9
Macrometastases	58	47.1
Patients with positive SLNs:	83	100
1 involved LN	52	62.7
>1 involved LNs	31	37.3
Tumor size:		
pT1	82	66.7
pT2	35	28.5
pT3	3	2.4
pT4	2	1.6
Lymph nodes status:		
pN0	41	33.3
pN1	70	56.9
pN2	8	6.5
pN3	4	3.2
Nottingham Histologic Grade:		
G1	21	17.1
G2	50	40.7
G3	51	41.5
Histologic type:		
NOS *	105	85.4
ILC **	15	12.2
Other	3	2.4
Hormone receptor status:		
Negative	16	13
Positive	102	83
HER2 status:		
Normal	92	74.8
Overexpression	26	21.1

* NOS—invasive carcinoma of no special type, ** ILC—invasive lobular carcinoma.

**Table 2 ijms-23-08461-t002:** Densities of investigated DC subpopulations in different compartments of tumor-free, micrometastatic, and macrometastatic SLNs.

		Tumor-Free	Micro-Metastatic	Macro-Metastatic	*p*-Value
CD1a	intratumoral	-	-	1.55 ± 2.09	-
	tumor margin	-	10.58 ± 10.38	18.74 ± 20.19	NS **
	distant area	63.25 ± 31.16	61.08 ± 33.79	46.76 ± 26.87	0.055 *
CD1c	intratumoral	-	-	0.45 ± 0.82	-
	tumor margin	-	13.28 ± 24.21	23.66 ± 22.73	0.031 **
	distant area	42.22 ± 34.64	49.13 ± 29.90	51.83 ± 38.52	NS *
DC-LAMP	intratumoral	-	0.70 ± 1.56	0.47 ± 1.19	-
	tumor margin	-	36.34 ± 36.46	68.69 ± 55.19	0.073 **
	distant area	143.09 ± 57.65	147.22 ± 57.71	136.11 ± 53.30	NS *
DC-SIGN	intratumoral	-	-	0.19 ± 0.42	-
	tumor margin	-	4.51 ± 6.55	16.58 ± 14.19	0.002 **
	distant area	18.71 ± 9.88	18.59 ± 9.73	22.45 ± 16.26	NS *

Mean values ± Standard deviation (SD). * Kruskal–Wallis ANOVA test. ** U Mann–Whitney test.

**Table 3 ijms-23-08461-t003:** Studies that investigated function of DC subsets and their role in tumors.

DC Marker	First Author, Date	Material	Conclusions	Reference
CD1a+	Studies investigating function of the DC subset			
	Gonçalves A.S., 2013	Cervical LNs from primary OSCC	Marker of immature DCs; Their accumulation in LNs associated with immunosuppressive microenvironment	[17]
	Cochran A.J., 2018	review	Expression of CD1a is not restricted to immature DCs exclusively	[2]
	Van de Ven R., 2011	SLNs from melanoma patients	CD1a+ DCs are poor activators of T cells; Contribute to immune tolerance even in their mature state	[18]
	Van de Ven R., 2012	DCs generated from their precursors	Marker of Langerhans cells	[19]
	Vermi W., 2003	Primary cutaneous melanoma patients (skin tumor & SLNs)	DCs exhibit capacity to coexpress molecules attributed to mature phenotype of LCs	[20]
	Thomachot M.C., 2004	Primary breast carcinoma	DCs with decreased ability to stimulate T cel proliferation; Immature DCs recruited to tumor site, where their maturation is impaired	[21]
	Studies investigating prognostic significance of the DC subset and its relationships with cancer progression			
	Gonçalves A.S., 2013	Cervical LNs from primary OSCC	Accumulation in LNs associated with occurrence of metastases	[17]
	La Rocca G., 2008	Primary invasive ductal breast carcinoma tumors and LNs	Accumulation of DCs associated with absence of nodal metastases	[22]
	Giorello M.B., 2021	Early invasive ductal breast carcinoma	Higher numbers of DCs associated with lower risk of metastatic disease	[23]
	Szpor J., 2021	Primary invasive breast cancer	Higher numbers of DCs associated with longer progression-free survival	[16]
	Kohrt H.E., 2005	LNs from breast cancer patients	Lower numbers of DCs associated with nodal metastases and recurrence	[24]
	Poindexter N.J., 2004 Blenman K.R.M., 2018 Mansfield A.S., 2011	SLNs from breast cancer patients	No relationship between DCs density and metastases in SLNs	[5,12,25]
CD1c+	Studies investigating function of the DC subset			
	Adams E.J., 2013	review	Molecule expressed on LCs	[26]
	Bourdely P., 2020	DCs progenitors from human blood	Inflammatory CD1c+ DC subset is distributed in numerous tissues and solid tumors; DCs capable of stimulate cytotoxic and helper T cells	[27]
	Tang-Huau T.L., 2018	Peritoneal ascites from cancer patients	DCs with capacity of stimulating effector cytotoxic T cells	[28]
	Kassianos A.J., 2012	Human blood DCs	CD1c+ DCs produce immunosuppressive and regulatory factors as well as exhibit a tolerogenic phenotype in response to bacterial stimulation	[29]
	Lavin Y., 2017	NSCLC tumor tissues and blood samples	Represent monocyte-derived DCs population generated at the tumor site	[30]
	Zekri A.R.N., 2018	Blood samples from chronic liver disease patients	Marker expressed on myeloid DCs producing IL-12	[31]
	Tabarkiewicz J., 2008	Tumor tissue, draining LNs and blood samples from NSCLC patients	CD1c+ DCs are trapped and accumulate at the tumor site	[11]
	Studies investigating prognostic significance of the DC subset and its relationships with cancer progression			
	Lavin Y., 2017	NSCLC tumor tissues and blood samples	More abundant in tumor tissue than in normal	[30]
	Zekri A.R.N., 2018	Blood samples from chronic liver disease patients	Less abundant in HCC patients than in normal	[31]
	Dyduch G., 2017	Cutaneous samples	Less abundant epidermal DCs in pre- and invasive melanoma than in bening nevi	[32]
	Tabarkiewicz J., 2008	Tumor tissue, draining LNs and blood samples from NSCLC patients	Higher DC numbers in tumors associated with worse survival	[11]
DC-LAMP	Studies investigating function of the DC subset			
	Cochran A.J., 2018	review	Marker of DC maturity	[2]
	Vermi W., 2003	Primary cutaneous melanoma patients (skin tumor & SLNs)	Marker of mature dermal DCs; Mutually exclusive expression with DC-SIGN; Mature DCs at tumor site show impaired ability to stimulate T cells	[20]
	O’Donell R.K., 2007	LNs from primary OSCC patients	Represent mature DCs;	[33]
	Movassagh M., 2004	Melanoma-positive SLNs	Mature DCs are pivotal contributors to melanoma immunosurveillance at the initial site of tumor spread	[34]
	Mansfield A.S., 2011	SLNs from breast cancer patients	In BC maturation and antigen presentation of DCs are arrested in SLNs	[25]
	Studies investigating prognostic significance of the DC subset and its relationships with cancer progression			
	Elliott B., 2007	Melanoma containing SLNs	Higher numbers of mature DCs in SLNs associated with longer survival and with antimetastatic immune response	[35]
	Movassagh M., 2004	Melanoma-positive SLNs	Higher numbers associated with occurence of tumor-free non-SLNs	[34]
	O’Donell R.K., 2007	LNs from primary OSCC patients	Represent mature DCs; Higher numbers in LNs associated with absence of metasases	[33]
	Mansfield A.S., 2011	SLNs from breast cancer patients	More dense infiltration related to absence of nodal metastases	[25]
DC-SIGN	Studies investigating function of the DC subset			
	Vermi W., 2003	Primary cutaneous melanoma patients (skin tumor and SLNs)	Marker expressed on immature dermal LCs; Mutually exclusive expression with DC-LAMP	[20]
	O’Donell R.K., 2007	LNs from primary OSCC patients	DC-SIGN+ DCs represent immature DCs with impaired antigen capture	[33]
	Zhou T., 2006	review	Marker expressed both on mature and immature DCs in dermis and mucosa; Expression of DC-SIGN on DCs contribute to tumor immune escape	[36]
	Van de Ven R., 2012	DCs generated from their precursors	Expression on interstitial DCs	[19]
	Deluce-Kakwata-nkor N., 2018	Monocyte-derived DCs from human blood samples	Expression on monocyte-derived DCs; After stimulation with LPS support pro-tolerant microenvironment	[37]
	Hossain M.K., 2019	review	Expression attributed primarily to dermal DCs; Represent immature and mature subsets in peripheral tissues and lymphoid organs, respectively	[7]
	Van de Ven R., 2011	SLNs from melanoma patients	Lower expression in SLN DCs attributed to maturation and migration of DCs	[18]
	Domínguez-Soto A., 2011	Monocytes from human blood samples	Expression observed on tumor-associated pro-tolerant macrophages; DC-SIGN+ cells present in stroma of several carcinoma tissues	[38]
	Merlotti A., 2019	Breast tumor and juxtatumoral samples	Expression observed on tumor-associated macrophages	[39]
	Spary L.K., 2014	Primary prostate cancer, prostate cancer cel lines and human blood samples	DCs represent immunosuppressive subset induced by stromal factors and cancer cells;	[40]
	Jubb A.M., 2010	Primary breast adenocarcinoma tissues	Expression on immature myeloid DCs	[41]
	Ammar A., 2011	Primary invasive breast cancer tissues	Marker of immature DCs	[42]
	Studies investigating prognostic significance of the DC subset and its relationships with cancer progression			
	O’Donell R.K., 2007	LNs from primary OSCC patients	Presence in primary tumor associated with poor survival	[33]
	Domínguez-Soto A., 2011	Monocytes from human blood samples	Interplay of DC-SIGN+ and cancer cells contribute to cancer progression	[38]
	Merlotti A., 2019	Breast tumor and juxtatumoral samples	Interaction between DC-SIGN+ macrophages and cancer cells contribute to cancer progression	[39]
	Jubb A.M., 2010	Primary breast adenocarcinoma tissues	Immature DCs related to worse survival	[41]
	Ammar A., 2011	Primary invasive breast cancer tissues	Immature DCs related to early recurrence	[42]

**Table 4 ijms-23-08461-t004:** Antibodies used in the study.

	Clone	Dilution	Antigen Retrieval	Incubation Time	Manufacturer
CD1a	MTB1	1:10	Citrate (40 min)	overnight	Novocastra (Leica Biosystems, Deer Park, IL, USA)
CD1c	5B8	1:200	EDTA (30 min)	30 min	Abcam (Cambridge, UK)
DC-LAMP	Rabbit polyclonal	1:50	EDTA (30 min)	30 min	Novus Bilogicals (Centennial, CO, USA)
DC-SIGN	5D7	1:50	EDTA (30 min)	30 min	Abcam (Cambridge, UK)
ER	6F11	1:100	Citrate (40 min)	30 min	Novocastra (Leica Biosystems, Deer Park, IL, USA)
PR	PgR636	1:100	Citrate (40 min)	60 min	Dako (Agilent Technologies, Santa Clara, CA, USA)
Ki67	MIB-1	1:100	Citrate (40 min)	30 min	Dako (Agilent Technologies, Santa Clara, CA, USA)

## Data Availability

The data presented in this study are available on request from the corresponding author. The data are not publicly available due to privacy restrictions.

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
