# Peer review of "Presence of Dendritic Cell Subsets in Sentinel Nodes of Breast Cancer Patients Is Related to Nodal Burden"

_ijms, 2022, doi:10.3390/ijms23158461_

Round 1
Reviewer 1 Report
The authors reported that presence of dendritic cell (DC) subsets in sentinel nodes of breast cancer (BC) patients is related to nodal burden. It is also reported that These relationships appear to be dependent on DC maturation and the characteristics of the BC.
The paper could be improved by addressing the following points.
Major comments;
1) A subset of DCs are evaluated by CD1a, CD1c, DC-LAMP, and DC-SIGN, but it is recommended to state in the introduction or discussion that the classification by these four is appropriate.
2) Is there a correlation between DC status in the sentinel lymph node and recurrence or long-term prognosis?
Reviewer 2 Report
LINE 88: "The patients who received neoadjuvant chemotherapy were excluded from the study". BUT THE SERIES INCLUDES AROUND 32% OF PATIENTS WITH pT2-4 TUMORS (cN NOT REPORTED). IT WOULD BE CONVENIENT TO EXPOSE YOUR NEOADJUVANT THERAPY INCLUSION CRITERIA.
TABLE 2. "NOS" OR "ILC" ARE ACRONYMS THAT COULD BE EXPLAINED AT THE BOTTOM OF THE TABLE. PERHAPS NOT ALL THE READERSHIP IS USED TO THEM.
Reviewer 3 Report
Materials: source of materials not clear
Was it from i) archives? (How many slides and from how many patients)
ii) New patients? (How many patients; age range)
iii) Or both i) and ii)
2.2. “Detection of SLNs and identification of nodal metastasis”
Lymphatic mapping……………………àisotope technique. (“Reference” is needed)
In general the experimental procedures were appropriately selected and accurately executed.
The analysis of the data generated was statistically extensive (good Figs and Tables), inclusive and sequentially written.
Summary: “To sum up, we conclude that CD1a+ DCs show protective activity against cancer progression while CD1c+, DC-LAMP+ and DC-SIGN+ subsets favor tumor spread. Moreover, the balance between these populations can be at least partially dependent on biologic features of tumors”.
In this summary, the link between DCs and the SLNs in the prognostication of BC is lost/missing
